# Cancer Development and Progression Through a Vicious Cycle of DNA Damage and Inflammation

**DOI:** 10.3390/ijms26073352

**Published:** 2025-04-03

**Authors:** Shosuke Kawanishi, Guifeng Wang, Ning Ma, Mariko Murata

**Affiliations:** 1Department of Pharmaceutical Sciences, Suzuka University of Medical Science, Suzuka 513-8670, Mie, Japan; 2Department of Acupuncture and Moxibustion Medical Science, Suzuka University of Medical Science, Suzuka 510-0293, Mie, Japan; wang@suzuka-u.ac.jp; 3Graduate School of Health Science, Suzuka University of Medical Science, Suzuka 510-0293, Mie, Japan; maning@suzuka-u.ac.jp; 4Institute of Traditional Chinese Medicine, Suzuka University of Medical Science, Suzuka 510-0293, Mie, Japan; 5Department of Environmental and Molecular Medicine, Mie University Graduate School of Medicine, Tsu 514-8507, Mie, Japan

**Keywords:** chronic inflammation, cancer, HMGB1, DNA damage

## Abstract

Infections and chronic inflammation play a crucial role in the development of cancer. During inflammatory processes, reactive oxygen and nitrogen species are generated by both inflammatory and epithelial cells, leading to the induction of oxidative and nitrative DNA damage, such as the formation of 8-oxo-7,8-dihydro-2′-deoxyguanosine (8-oxodG) and 8-nitroguanine (8-nitroG). These DNA alterations can trigger mutations, which are believed to contribute to cancer formation driven by inflammation. The authors observed the generation of 8-nitroG through iNOS expression in human and animal tissues under inflammatory conditions, where cancer is likely to develop. 8-NitroG serves as a predictive and prognostic indicator for cancers linked to inflammation. Inflammation causes DNA damage, and the subsequent DNA damage response can create an inflammatory environment marked by hypoxia, with HMGB1 being a key factor. The interplay between HIF-1α, NF-ĸB, and HMGB1 sustains DNA damage and the accumulation of mutations, driving cancer progression and worsening prognosis. 8-NitroG is involved not only in the onset and advancement of cancer but also in its progression and conversion. Herein, the authors propose a vicious cycle of DNA damage and inflammation in cancer development (initiation and promotion) and progression, including conversion, via HMGB1.

## 1. Introduction

Persistent infections, environmental factors, and inflammation-related disorders can trigger inflammatory responses. Long-term inflammation is implicated in around 25% of all human cancers [1,2]. During prolonged inflammation, the overproduction of reactive nitrogen species (RNS) by inducible nitric oxide synthase (iNOS) may be a critical factor in cancer development, primarily through DNA damage, notably via inflammation-specific 8-nitroG. NF-ĸB is instrumental in influencing the release of inflammatory factors such as TNF-α, IL-6, and IL-1β, which are key in the regulation of carcinogenesis [3]. This result indicates that 8-nitroG holds potential as a biomarker for assessing the risk of cancer development linked to inflammation [4]. Kawanishi et al. established a method to produce a highly sensitive and specific anti-8-nitroG rabbit polyclonal antibody and detect 8-nitroG formation in biopsy and surgical specimens and animal tissues using immunohistochemistry [5].

Under inflammatory conditions, genetic/epigenetic alterations occur in normal cells, contributing to driver gene mutation, promoter methylation of tumor suppressor genes, microRNA dysregulation, etc. [2]. Genetic variations influence an individual’s vulnerability to stress induced by carcinogens. An imbalance in the microbiota can heighten the likelihood of cancer, particularly colorectal cancer. Mechanisms that allow tumors to evade the immune system may also impact the relationship between inflammation and DNA damage. While there are other potential ways that interactions between the host and microbes, genetic variations, and immune evasion strategies may affect this relationship, our focus is on how inflammation-driven carcinogenesis begins with DNA damage, notably 8-nitroG, and progresses through a vicious cycle involving HMGB1.

## 2. DNA Damage in Inflammation-Related Cancer

### 2.1. Pathogens

#### 2.1.1. *Opisthorchis viverrini*

*Opisthorchis viverrini (OV)* infection is linked to an elevated risk of cholangiocarcinoma (CCA), potentially via persistent inflammation. OV-induced CCA serves as a typical example of cancer development driven by inflammatory processes. 8-NitroG has been identified in human CCA samples [6,7]. Research by Pinlaor et al. [8] highlighted the significance of multiple OV infections in hamsters. The levels of 8-nitroG and 8-oxodG, along with iNOS expression, seemed to rise in the bile duct epithelium following three infections, then two, and finally a single infection after the number of inflammatory cells decreased. 8-NitroG and 8-oxodG accumulate in the bile duct epithelial cells with repeated OV infection. Inflammatory cells are recruited in acute phase, and their levels decrease thereafter. On the other hand, iNOS expression continues in the bile duct epithelial cells, since TLRs, which participate in iNOS induction, are expressed in only not immune cells but also epithelial cells. Pinlaor et al. demonstrated that the anthelminthic drug praziquantel can exert a preventive effect on OV-induced CCA in hamsters by inhibiting iNOS-dependent DNA damage through not only the elimination of parasites but also a potential anti-inflammatory effect [9].

Pinlaor and colleagues [6] conducted immunohistochemical studies, which showed that HIF-1α was present in the malignant tissues of all patients, indicating hypoxic conditions within the tumors. The expression of HIF-1α was associated with iNOS levels and 8-oxodG formation in the cancerous tissue, hinting at a connection between iNOS and HIF-1α.

#### 2.1.2. *Schistosoma haematobium*

Schistosomiasis is the leading helminth infection globally, causing significant morbidity and mortality. The majority of cases occur in Africa, with approximately two-thirds attributed to *Schistosoma haematobium* (SH) [10]. One pathway to cancer involves the parasitic eggs, which embed in the bladder, causing irritation, fibrosis, chronic cystitis, and, ultimately, cancer. SH, an inflammatory parasite, is known to be carcinogenic and can lead to squamous cell and urothelial carcinomas of the bladder.

Ma et al. [11] conducted an immunohistochemical analysis to assess the presence of 8-nitroG and 8-oxodG in bladder samples from individuals with cystitis and bladder cancer, both infected with SH. Additionally, they evaluated the expression of NF-κB and iNOS, which contribute to the formation of 8-nitroG. The staining levels for 8-nitroG and 8-oxodG were notably elevated in bladder cancer and cystitis tissues compared to those in normal tissues. In bladder cancer patients, iNOS was found to be co-localized with NF-κB in cells that tested positive for 8-nitroG.

#### 2.1.3. *Helicobacter pylori*

*Helicobacter pylori* (*H. pylori*) infection of the gastric mucosa leads to active chronic gastritis, gastroduodenal ulcers, and MALT lymphoma, laying the groundwork for understanding the general relationship between chronic infection, inflammation, and cancer [12,13].

A dual immunofluorescence labeling investigation [14] revealed that the amounts of 8-nitroG and 8-oxodG in the gastric gland epithelium were markedly elevated in individuals with *H. pylori*-induced gastritis compared to those in individuals without the infection. In patients with *H. pylori*, there was a notable increase in proliferating cell nuclear antigen (PCNA), which collaborates with DNA polymerase to enhance DNA synthesis [15], in the gastric gland epithelial cells. This accumulation of PCNA was strongly associated with the presence of 8-nitroG and 8-oxodG.

Katsurahara and colleagues [16] noted a strong correlation between the generation of 8-oxodG and 8-nitroG and the extent of glandular atrophy, the presence of chronic inflammatory cells, and the occurrence of intestinal metaplasia in the glandular epithelial cells of the corpus mucosa in individuals with *H. pylori*-positive gastritis. After the successful eradication of *H. pylori*, there was a notable decrease in the infiltration of chronic inflammatory cells and neutrophilic activity, as well as a reduction in the levels of 8-oxodG and 8-nitroG.

#### 2.1.4. Hepatitis B/C Virus

Hepatitis B virus (HBV) and hepatitis C virus (HCV) are the primary causes of chronic hepatitis and progressive liver fibrosis, which can lead to cirrhosis and hepatocellular carcinoma (HCC). HCC is among the most prevalent cancers globally, and its mortality rate has risen over the past few decades.

Horiike et al. [17] assessed the production of 8-nitroG and 8-oxodG in the livers of individuals with chronic hepatitis C before and after interferon treatment. Immunohistochemical (IHC) analysis revealed strong reactivity for 8-nitroG and 8-oxodG in the livers of patients with chronic hepatitis C but not in control livers. 8-NitroG was found to build up in hepatocytes, especially in the periportal region. The levels of 8-nitroG and 8-oxodG increased with higher grades of inflammation. In patients who responded to treatment, the accumulation of 8-nitroG and 8-oxodG significantly decreased in the liver following interferon therapy.

To assess sequential shifts in gene expression patterns in a mouse model of chronic immune-mediated hepatitis, Nosaka et al. [18] conducted studies using HBV-transgenic mice. IHC analysis revealed that the buildup of 8-nitroG in liver cells and their regenerative growth intensified as the chronic liver condition advanced.

#### 2.1.5. Human Papillomavirus

Research in molecular epidemiology has demonstrated that certain strains of human papillomavirus (HPV) are linked to cervical cancer [19]. It is suggested that inflammation significantly contributes to the progression of HPV-related cervical cancer [20].

Hiraku and colleagues [21] identified the presence of 8-nitroG and 8-oxodG in biopsy samples and compared these with the levels of the cyclin-dependent kinase inhibitor p16, a known biomarker for cervical neoplasia. The immunoreactivity of 8-nitroG and 8-oxodG showed a significant correlation with the grade of cervical intraepithelial neoplasia (CIN). PCNA is specifically expressed in dysplastic epithelial cells, whereas it is absent in benign condyloma acuminatum. No statistically significant differences in p16 expression were found between CIN and condyloma acuminatum samples. High-risk HPV types are known to induce iNOS-dependent DNA damage, contributing to dysplastic changes and carcinogenesis. Conversely, p16 seems to be an indicator of HPV infection. Thus, 8-nitroG appears to be a more appropriate biomarker for assessing inflammation-driven cervical carcinogenesis compared to p16.

#### 2.1.6. Epstein–Barr Virus

Significant progress has been made in comprehending nasopharyngeal carcinoma (NPC), a cancer linked to Epstein–Barr virus (EBV) and prevalent in regions such as Southern China, Southeast Asia, and North Africa [22].

Ma et al. [23] collected biopsy and surgical samples of nasopharyngeal tissues from NPC patients in Southern China. They utilized double immunofluorescent staining to reveal the significant presence of 8-nitroG and 8-oxodG in both cancerous and inflammatory cells within the stroma of NPC tissues. Notably, strong iNOS reactivity was identified in the cytoplasm of 8-nitroG-positive cancer cells. In EBV-positive patients with chronic nasopharyngitis, DNA damage and iNOS expression were also detected in epithelial cells, albeit at much lower levels compared to those in NPC patients. The cancer cells of NPC patients showed high levels of EGFR and phosphorylated STAT3, indicating that a STAT3-dependent pathway is crucial for NPC development. IL-6 was primarily found in the inflammatory cells of EBV-infected nasopharyngeal tissues. In vitro, LMP1-expressing cells exhibited the nuclear accumulation of EGFR, and IL-6 induced the phosphorylation of STAT3 and iNOS. These findings imply that the nuclear buildup of EGFR and the activation of STAT3 by IL-6 are essential for iNOS expression and the resulting DNA damage, which contribute to the progression of EBV-associated NPC.

### 2.2. Environmental Factors

#### 2.2.1. Ultraviolet Light

Ultraviolet (UV) radiation from the sun is recognized as an environmental carcinogen affecting humans. The detrimental impacts of UV exposure, whether from natural sunlight or artificial therapeutic lamps, pose significant risks to human well-being. According to Yu et al. [24], when normal human skin is exposed to UV radiation, it can lead to immediate inflammatory responses like sunburn (erythema) and molecular-level DNA damage, such as cyclobutane pyrimidine dimers and (6-4) photoproducts.

Ma et al. [25] demonstrated that 8-nitroG and 8-oxodG were formed in epidermal cells, especially in basal cells in skin tissues of UV (UVB, 280–360 nm)-exposed hairless mice. In the dermis, the presence of macrophages and iNOS expression has been noted. As shown in Figure 1, 8-nitroG was weakly expressed in epidermal cells at 2.5 months and moderately at 3.5 months. After five months exposure, the strong staining of 8-nitroG and atypia of basal keratinocytes with a loss of polarization, crowding, and overlapping were observed; this condition, often diagnosed as “actinic keratosis” in humans, is considered the most frequent precursor to cutaneous squamous cell carcinoma. It has been suggested that inflammation-related 8-nitroG formation contributes to skin carcinogenesis in addition to UV-specific DNA damage, such as pyrimidine dimers and (6-4) photoproducts.

Superoxide dismutase (SOD) is recognized for its protective role against oxidative stress-induced skin issues. Shariev et al. [26] explored the therapeutic potential of RM191A, a new SOD mimetic, in a topical gel using a human skin explant model and observed that it significantly inhibited UV-induced DNA damage in the epidermis and dermis, as indicated by levels of cyclobutane pyrimidine dimers, 8-oxoG, and 8-nitroG.

#### 2.2.2. Asbestos and Nanomaterials

Asbestos causes lung cancer and malignant mesothelioma. One mechanism of carcinogenesis involves the induction of chronic inflammation. Some carbon nanotubes are fibrous, similar to asbestos. Nanomaterials, such as indium compounds, may also contribute to chronic inflammation similar to that caused by a foreign body, in addition to their individual toxic properties.

Hiraku et al. [27] investigated the connection between 8-nitroG formation and asbestos levels in human lung tissues. In individuals without mesothelioma, the staining intensities of 8-nitroG were notably linked to the total asbestos and amphibole content but not to chrysotile levels. This research is the first to show that 8-nitroG formation is related to asbestos presence in human lung tissues, suggesting that 8-nitroG could be a biomarker for assessing asbestos exposure and cancer risk.

Carbon nanotubes (CNTs) are widely recognized for their potential industrial applications in material science due to their distinctive physicochemical properties. Nevertheless, certain studies have indicated that CNTs can cause mesothelioma in animals. Hiraku et al. [28] first demonstrated a carcinogenic mechanism indicating that toll-like receptor (TLR) 9 participates in multi-walled CNT-induced genotoxicity and that nitrative DNA damage may contribute to carcinogenesis.

Indium compounds play a role in the production of screens for smartphones and TVs. These substances can lead to interstitial pneumonia among those who come into contact with them. Studies on animals indicate that indium compounds may trigger lung cancer. In their research, Ahmed et al. [29] indicated that both indium nanoparticles and indium ions significantly increased 8-nitroG formation in A549 human lung alveolar epithelial cells, and this effect was largely suppressed by siRNAs targeting HMGB1; receptor for advanced glycation end products (RAGE) and TLR9 were involved. Persistent inflammation is believed to contribute to lung cancer development and fibrosis caused by asbestos and nanomaterials.

### 2.3. Inflammation-Related Disorders

#### 2.3.1. Oral Lichen Planus/Leukoplakia

Oral lichen planus (OLP) is a persistent inflammatory condition of the mucous membranes linked to oral cancer. A dual immunofluorescence labeling study [30] revealed that 8-nitroG and 8-oxodG accumulate in the oral epithelial tissue of OLP and oral squamous cell carcinoma (OSCC) samples, with minimal or no reactivity in healthy oral mucosa. The study also noted the co-occurrence of 8-nitroG and iNOS in the oral epithelia of both OLP and OSCC. The accumulation of p53 was more pronounced in the oral epithelium of OSCC compared to OLP, and no p53 buildup was detected in normal oral mucosa. This suggests that DNA damage mediated by iNOS may contribute to the accumulation of p53 in OLP and OSCC.

Oral leukoplakia, a condition that can precede oral cancer, has been linked to specific histological alterations. Ma et al. [31] noted the presence of epithelial dysplasia and inflammatory cell infiltration in the oral tissues of individuals with leukoplakia. Using double immunofluorescence labeling, they found that mutagenic 8-nitroG and 8-oxodG accumulate in the oral epithelia of these patients, while the normal oral mucosa shows minimal or no such reactivity. Additionally, the expression of iNOS is detectable in the oral epithelia of those with leukoplakia. Furthermore, PCNA is present in the basal layer’s 8-nitroG-positive epithelial cells.

#### 2.3.2. Inflammatory Bowel Disease

Colorectal cancer is often a consequence of prolonged colon inflammation, such as that seen in inflammatory bowel disease (IBD). Diagnosing IBD through histopathology involves recognizing a persistent inflammatory pattern with specific spatial distribution, marked by structural irregularities in the intestinal lining and a distinctive cellular makeup of the inflammatory cells [32].

To investigate the potential role of iNOS-induced DNA damage in IBD-related carcinogenesis, Ding et al. [33] developed an IBD mouse model. In this model, 8-nitroG and 8-oxodG were predominantly detected in the epithelial cells. 8-NitroG is formed in most 8-oxodG-positive nuclei of epithelial cells, and iNOS was found in the colonic epithelium. iNOS-mediated DNA damage in the colon epithelial cells, followed by cell proliferation, may contribute to the development of colon cancer.

Ulcerative colitis (UC) is characterized by a diffuse pattern of inflammation and usually affects the rectum with variable extension toward the terminal ileum. Prolonged UC duration elevates the likelihood of developing cancers linked to UC. Research by Saigusa et al. [34] indicated that 8-nitroG levels were notably higher in the group with UC-related carcinoma compared to the non-carcinoma group. PCNA is present in the colonic epithelium. The occurrence of 8-nitroG is more prevalent in individuals with UC-associated cancer than in those with sporadic colorectal cancer.

#### 2.3.3. Gastric Acid Reflux (Barrett’s Esophagus)

Barrett’s esophagus (BE), a condition linked to stomach acid reflux, increases the likelihood of developing Barrett’s esophageal adenocarcinoma (BEA) [35]. The use of proton pump inhibitors (PPIs) in BE patients is anticipated to lower the risk of BEA.

Thanan and colleagues [36] conducted an immunohistochemical analysis to evaluate the presence of 8-nitroG and 8-oxodG in normal esophageal tissue, Barrett’s esophagus (BE), and BE-associated adenocarcinoma (BEA), also BE before and after PPI therapy. The study revealed that DNA damage was most pronounced in BEA, followed by BE, and then normal tissue. Similarly, iNOS levels were highest in BEA, intermediate in BE, and lowest in normal tissue. In contrast, Mn-SOD expression was lowest in BEA, higher in BE, and highest in normal tissue. Following 3–6 months of PPI treatment, there was a notable rise in Mn-SOD expression and Nrf2 nuclear localization, along with a reduction in DNA damage in BE tissue. 8-NitroG and 8-oxodG are critical in the development of BEA from BE. PPIs may activate and promote the nuclear translocation of Nrf2, which in turn enhances Mn-SOD expression and mitigates DNA damage.

#### 2.3.4. Malignant Fibrous Histiocytomas (Undifferentiated Pleomorphic Sarcomas)

Malignant fibrous histiocytoma (MFH), a frequently occurring soft tissue sarcoma in adults, is known for its grim prognosis. While various factors have been suggested as potential causes of MFH, there is evidence pointing to the involvement of inflammatory processes. Hoki et al. [37] found that the formation of 8-nitroG and 8-oxodG was significantly more pronounced in MFH tissues compared to the adjacent non-tumor tissues. In these tissues, iNOS, NF-ĸB, and COX-2 were observed to be co-located with 8-nitroG. A Kaplan–Meier analysis indicated that a higher intensity of 8-nitroG staining was correlated with a worse outcome. Hoki et al. [38] analyzed the presence of 8-nitroG and HIF-1α in clinical samples from MFH patients. They found that 8-nitroG was mainly concentrated in the nuclei of cancer cells, while HIF-1α was present in both the cytoplasm and nuclei. A Kaplan–Meier survival analysis revealed a significant difference between groups with high and low levels of 8-nitroG (*p* = 0.00003) and HIF-1α (*p* = 0.01104). These findings indicate that the iNOS-dependent formation of 8-nitroG, influenced by HIF-1α and NF-ĸB/iNOS, is crucial in the development of inflammation-associated cancers.

### 2.4. 8-NitroG as a Biomarker Candidate

We compiled the key method data from the most frequently cited studies in Appendix A. Table 1 shows a comparison of biomarkers identified by IHC in cited preclinical studies. 8-NitroG and 8-oxodG in urine, serum/plasma, and tissue samples can be detected by IHC and other techniques, such as HPLC with electro-chemical detection, LC-MS/MS, and ELISA. Methods with the higher detectable sensitivity and lower cost should be established to apply 8-nitroG in clinical settings. The findings are from preclinical studies, and therefore, longitudinal human studies are required to validate the sensitivity/specificity of 8-nitroG or other biomarkers to detect target diseases.

## 3. DNA Damage and Cancer Stem Cells

Cancer stem cells (CSCs) exhibit properties of self-renewal, differentiation, and the ability to form tumors when introduced to an animal host. Additionally, CSCs are involved in tumor initiation, progression, metastasis, recurrence, and resistance to treatment.

Nitrative and oxidative DNA damage in stem cells may be crucial in the onset of inflammation-driven carcinogenesis. Since cancer likely originates from DNA mutations that accumulate via normal stem cell division, inflammation may promote cancer development even before stem cells become cancerous [39]. Inflammation might also play a role in the development, maintenance, and proliferation of CSCs. We observed 8-nitroG formation in stem cell marker-positive cells in inflammatory tissues at the site of carcinogenesis. CSCs are crucial for tumor initiation due to the ability to self-renew, differentiate, avoid immune destruction, and promote inflammation and angiogenesis [40].

CSCs represent a subpopulation within tumors and promote cancer progression, metastasis, and recurrence owing to their self-renewal capacity and resistance to conventional therapies [41]. As described below, 8-nitroG and 8-oxodG are formed via iNOS expression in CSC marker-positive cells in patients with various types of cancers. Within the inflammatory tumor microenvironment, inflammatory agents can trigger the epithelial–mesenchymal transition (EMT) [42]. CSCs facilitate cancer progression in such an environment through DNA damage, genetic mutations, and chromosomal instability [43].

### 3.1. CSCs in OV-Induced Cholangiocarcinoma

Thanan and colleagues [7] investigated multiple CSC markers (including CD133, CD44, OV6, and Oct3/4), 8-nitroG, 8-oxodG, DNA damage response (DDR) proteins, and Mn-SOD in CCA samples. The quantitative assessment of 8-oxodG indicated markedly elevated levels in tumor tissues positive for CD133 and/or Oct3/4 compared to those in tissues negative for these markers. Patients with CD133- and/or Oct3/4-positive CCA exhibited significantly reduced Mn-SOD expression and increased gamma-H2AX, a DDR protein. The presence of CD133 and/or Oct3/4 in CCA was strongly linked to a worse prognosis. These markers in CCA are associated with heightened DNA damage and DDR proteins, which may contribute to genetic instability, thereby promoting cancer progression and more aggressive clinical outcomes.

Multiple CD44 isoforms are present in various cancer stem cells (CSCs) during tumor development and metastasis. Specifically, CD44 variant 9 (CD44v9) is notably abundant in cancers triggered by chronic inflammation. Suwannakul et al. [44] reported that CD44v9 levels were higher in CCA tissues infected with OV compared to those in non-infected CCA tissues. They also noted a significantly greater expression of CD44v9 in OV-CCA tissues relative to both non-OV-CCA and normal liver tissues. In another study, Suwannakul et al. [45] explored the potential functions of CD44v9 in cultured human CCA cells. They found that the mRNA levels of CD44v9 were markedly increased in these cells. When CCA cells were treated with CD44v9 siRNA, their migration and invasion capabilities were reduced. Additionally, downregulating CD44v9 led to a decrease in CCA tumor growth in mouse xenograft models. The knockdown of CD44v9 also resulted in a reduction in epithelial–mesenchymal transition (EMT). These findings suggest that CD44v9 may serve as a new marker for CCA stem cells, potentially linked to the prognosis of inflammation-related cancers.

### 3.2. CSCs in SH-Infected Bladder Cancer Cells

Ma et al. [11] observed a notable rise in Oct3/4 expression in cells from SH-related bladder cancer tissues compared to that in normal bladder and non-infected cancer tissues. Additionally, 8-nitroG was detected in Oct3/4-positive stem cells within SH-related cystitis and cancer tissues. The inflammatory response triggered by SH infection might lead to an increase in the number of mutant stem cells, where DNA damage occurs through iNOS activation, mediated by NF-κB, thus initiating the carcinogenic process.

Inflammation can potentially trigger stem cells through the generation of prostaglandin (PG) E2, which is driven by COX-2. Thanan et al. [46] conducted an immunohistochemical study on the expression of stem cell markers (Oct3/4 and CD44v6) and COX-2 in bladder tissue samples from individuals with cystitis and cancer. The findings indicated a strong association between the presence of these stem cell markers and the nuclear positioning of COX-2, implying that inflammation might influence the multiplication and reprogramming/differentiation of stem cells.

### 3.3. CSCs in Nasopharyngeal Carcinoma

Wang et al. [47] investigated multiple markers of cancer stem/progenitor cells (CD44v6, CD24, and ALDH1A1) in NPC tissues and cell lines through dual immunofluorescence staining. The 8-nitroG staining intensity was notably higher in cancer and inflammatory cells within NPC tissues compared to that in chronic nasopharyngitis tissues. In primary NPC samples, the expression of CD44v6 and ALDH1A1 was significantly elevated in cancer cells relative to that in chronic nasopharyngitis tissues. Likewise, the NPC cell line exhibited more pronounced staining for CD44v6 and ALDH1A1 than the immortalized nasopharyngeal epithelial cell line. 8-NitroG accumulation was observed in both CD44v6- and ALDH1A1-positive stem cells in NPC tissues. The increased formation of DNA damage due to inflammation may lead to mutations in stem cells, potentially driving the development of NPC.

### 3.4. CSCs in Barrett’s Esophageal Adenocarcinoma

Thanan et al. [48] investigated the formation of 8-nitroG and 8-oxodG, along with the expression of CD133, in biopsy samples from patients with BE and BEA and in normal tissues. In BE and BEA tissues, CD133 was observed on the surface of columnar epithelial cells and in the cytoplasm and cell membrane of cancer cells in BEA tissues. CD133 was found to co-localize with DNA damage in both columnar epithelial and cancer cells. These findings indicate that in BE, columnar epithelial cells expressing CD133 on their apical surface may experience inflammation-induced DNA damage, and these mutated cells may develop characteristics of CSCs with CD133 expression.

## 4. Mechanism of Inflammation-Related Carcinogenesis

Growing evidence suggests that inflammation plays a significant role in the onset and advancement of cancer. Prolonged inflammation can transform healthy cells into cancerous ones and accelerate tumor growth. In an inflammatory environment, both inflammatory and epithelial cells produce reactive oxygen species (ROS) and reactive nitrogen species (RNS), which can result in DNA damage, including the formation of 8-oxodG and 8-nitroG. 8-NitroG is formed by infection, inflammatory diseases, and physicochemical factors at the sites where cancer will appear [1]. In addition, owing to constant exposure to carcinogens (DNA-damaging agents), cells may undergo DNA damage, which can lead to mutations, resulting in cancer. However, DDR corrects these potentially tumorigenic problems via DNA repair and cell death [49]. When DNA repair mechanisms falter, DDR pathways can lead to mutations and genomic instability, thereby triggering cancer and facilitating its malignant progression [50]. Hypoxia-inducible factors (HIFs) are involved in various aspects of tumor development, such as cell proliferation, blood vessel formation, metastasis, and resistance to drugs [51]. It has been reported that HMGB1 is upregulated in cancer tissue samples and is positively associated with poor prognosis in cancer patients in relation to HMGB1-induced DDR [52].

### 4.1. HMGB1 as a Key Factor in the Mechanism

Nuclear HMGB1 is a non-histone chromatin-associated protein with various biological functions, including DNA repair, transcription, replication, and genome stability under normal physiological conditions. HMGB1 exhibits some anticancer activities by promoting dendric cell maturation and activating the RAGE/NF-kB pathway to polarize M1-like macrophages [53]. On the other hand, HMGB1 is released from necrotic cells and actively secreted by immune cells under pathological inflammatory conditions such as hypoxia. During hypoxia, HMGB1 moves from the nucleus to the extracellular environment via an unclear mechanism and interacts with RAGE. This interaction triggers the activation of HIF-1α and NF-κB through the PI3K/Akt and RAS/RAF/MEK/ERK pathways [54,55]. Additionally, HMGB1 binds to TLRs, activating the MyD88/NEMO pathway [56]. This leads to the nuclear translocation of NF-κB, which then induces the production of proinflammatory cytokines IL-6 and TNF-α.

Ahmed et al. [29] showed that the HMGB1/RAGE/TLR9 axis played a role in triggering inflammation-induced DNA damage in lung epithelial cells exposed to indium compounds. As shown in Figure 2, AOM/DSS-induced colorectal cancer in mice was associated with high expression of both HMGB1 and RAGE in cancer tissues. Our findings indicate that glycyrrhizin (GL), a key component found in licorice root, hinders the development of cancer by suppressing inflammation in a mouse model of colorectal cancer, since GL weakens the proinflammatory effects of HMGB1 by specifically binding GL to HMGB1 [57,58]. Interestingly, recent reports have shown high expression of HMGB1 in cancer tissues compared to that in adjacent tissues [59] and high serum levels of HMGB1 in cancer patients with metastasis [60]. These reports suggest that the cycle between DNA damage and inflammation via the HMGB1/RAGE/TLR9 pathway plays an important role in cancer development and progression.

### 4.2. Cancer Development and Progression Occur Through a Vicious Cycle of DNA Damage and Inflammation via HMGB1

We envision a possible mechanism for cancer development and progression involving a vicious cycle of DNA damage and inflammation via HMGB1 (Figure 3). As shown in Figure 3a, carcinogenic infection and inflammatory factors induce inflammation in both inflammatory and epithelial cells. Consequently, ligands like foreign pathogen-associated molecular patterns (PAMPs) and internal damage-associated molecular patterns (DAMPs), such as HMGB1, attach to receptors, including TLRs and RAGE, triggering the NF-κB signaling pathway [61]. This leads to the secretion of proinflammatory cytokines, specifically TNF-α and IL-6, from immune cells like macrophages and neutrophils. TNF-α and IL-6 upregulate iNOS and NADPH oxidase (Nox) expression to generate NO and O_2_^−^, forming ONOO^−^, which can induce DNA damage, such as mutagenic DNA lesions and the formation of 8-nitroG and 8-oxodG. DNA damage triggers DNA repair mechanisms and can also lead to cell death through the DDR pathway. When cells die, HMGB1 is released and can re-engage with RAGE/TLRs. Conversely, if DDR fails, it can cause mutations and genomic instability, potentially leading to cancer. The interaction between HMGB1 and RAGE then enhances tumor cell proliferation by advancing the cell cycle (Cyclin D1, Cyclin E1, PCNA) through NF-κB pathways [54]. NF-κB also activates COX-2, which converts arachidonic acid (AA) into prostaglandins such as PGE2. PGE2 reacts with its receptor (EP2) to activate NF-κB, again resulting in an inflammatory microenvironment.

COX-2 and its downstream product, PGE2, are crucial in creating an inflammatory microenvironment in tumor tissues and maintaining stemness, as indicated by the presence of stem cell markers, including CD44 and SOX9. Our previous study [62] supported the importance of the COX-2/PGE2 pathway, showing that aspirin, a COX-2 inhibitor, significantly decreased the number and size of colorectal tumors and reduced TNF-α and ROS levels in the plasma of AOM/DSS-induced colorectal cancer model mice. It also led to a decrease in 8-nitroG formation and oncogenic YAP1, which can induce CSCs.

NF-ĸB is instrumental in altering the release of inflammatory factors such as TNF-α and IL-6, which are pivotal in the regulation of carcinogenesis. NF-ĸB upregulation can intensify inflammation, potentially leading to the development of cancer. NF-ĸB plays a role in controlling CSCs [3] and is involved in the accumulation of CSCs [63].

Exogenous IL-6 rescues the reduced sphere-forming capacity and proportion of CSCs [64]. The IL-6/JAK/STAT3 pathway is overactive in numerous cancer types. The IL-6/STAT3 pathway is essential for transforming non-CSCs into CSCs [65]. The expression of the IL-6 receptor enhances CSC self-renewal through STAT3 activation and the upregulation of key CSC factors such as c-MYC, KLF4, and SOX9. The IL-6/STAT3 signaling pathway is also necessary for EMT in CD133(+) and CD44(+) CSCs [66].

IL-6/YAP1 signaling, as well as IL-6/STAT3, participates in inflammation-driven regeneration [67]. The Hippo coactivator YAP1 regulates the tumor immune microenvironment through the IL-6/STAT3 signaling pathway [68]. Wang et al. [69] demonstrated that YAP1 enhances SOX2/9 expression, imparting stem-like characteristics to cancer cells. YAP1 controls the transcription of SOX2/9 via a conserved TEAD-binding motif within the SOX promoter.

Additionally, NO participates in the generation of CSCs. Chronic exposure to NO induces stem-cell phenotypes with malignant characteristics [70]. NO, a biological mediator often overexpressed in cancer, is associated with tumor aggressiveness.

Inflammatory processes lead to tissue damage through the action of ROS/RNS and cytokines. Stem and progenitor cells play a crucial role in preserving tissue balance and mending injuries [39]. Progenitor/stem cells proliferate to regenerate tissues. The ROS/RNS-rich environment causes mutations in stem cells to become cancerous. Thus, DNA damage/mutations in both normal epithelial cells and stem cells can participate in the initiation and promotion of cancer development [71].

The buildup of mutations within the initial cells triggers the malignant transition of cancer cells and tumor development (Figure 3b). As the tumor grows, it promotes a hypoxic environment, leading to the increased production of HIF-1α. The significance of HIF-1α in the advancement of cancer is well recognized [72]. Under hypoxic conditions, the interaction between tumor and immune cells activates the HIF-1α, NF-ĸB, and STAT pathways, enhancing the production and secretion of proinflammatory cytokines [73]. It is proposed that mutual activation between HIF-1α and NF-ĸB/iNOS contributes to ongoing DNA damage, potentially leading to increased tumor invasiveness through mutations and, ultimately, a worse prognosis [74].

HMGB1, a protein found in the nucleus, is discharged from cells that have died [56]. Nuclear HMGB1 translocates to the extracellular space under hypoxic conditions depending on tumor growth [55] and the inflammatory tumor microenvironment [56,75]. Extracellular HMGB1 binds to TLRs and RAGE and activates NF-κB signaling pathways [55,76], thereby inducing the proinflammatory cytokines TNF-α and IL-6, which escalate and prolong the inflammatory reaction. When NF-κB is overexpressed, it boosts the production of COX-2 and iNOS, resulting in DNA damage, which in turn triggers the continuous release of HMGB1.The repeated release of HMGB1 is a key factor in the formation of this vicious cycle between DNA damage and inflammation, which promotes cancer progression and conversion.

CSCs are generated during cancer development and progression. During cancer development, mutations can lead to the dedifferentiation of normal stem cells into CSCs in several processes, such as regeneration and wound healing during inflammation. Through the process of dedifferentiation, cells can acquire stem-like characteristics, including self-renewal and pluripotency [77]. In the progression of cancer, tumor-associated macrophages enhance the expression of genes related to stemness in cancer cells. Inflammatory factors (IL-6, STAT3, YAP1, NF-κB, COX-2/PGE2, and NO) induce stemness, which promotes cancer progression.

It is suggested that the harmful cycle of DNA damage and inflammation significantly influences cancer development and progression through HMGB1.

## Figures and Tables

**Figure 1 ijms-26-03352-f001:**
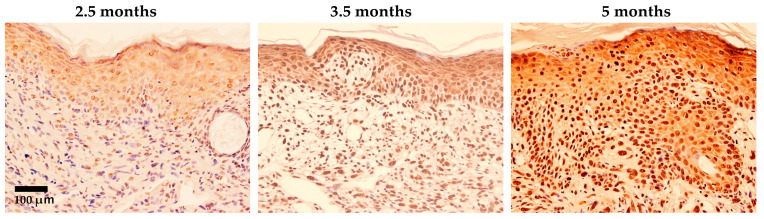
Immunohistochemical localization of 8-nitroG in UV-exposed mouse skin specimens (modified from the work of Ma et al. (2011) [25]). Hairless mice HOS (HR/De) were housed in a cage and exposed to UV light (280–360 nm, 6.3 J/m^2^/s, 3430 J/m^2^ at once) repeatedly twice per week for five months. Sections of mouse skin exposed to UV were stained with 8-nitroG antibodies. About 20% of the epidermal cells showed nuclear staining after 2.5 months. After 3.5 months and 5 months of UV exposure, strong nuclear staining was observed in roughly 70% and 100% of epidermal cells, respectively. Hematoxylin was used for counterstaining. The original magnification was 200×.

**Figure 2 ijms-26-03352-f002:**
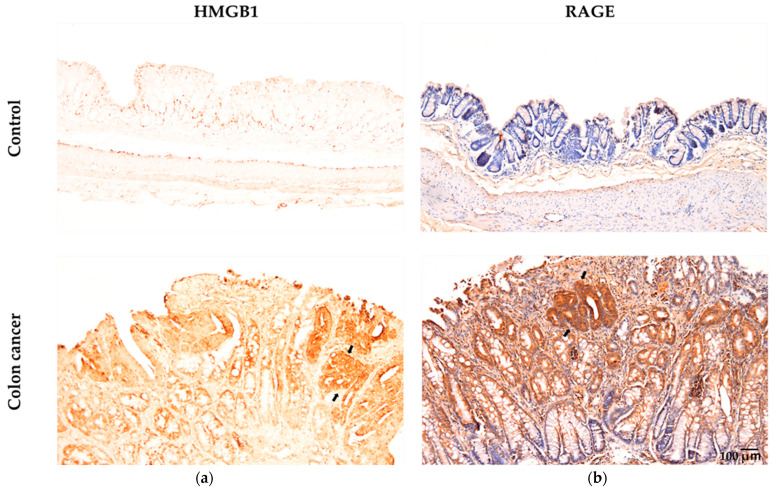
Immunohistochemical examination of HMGB1 and RAGE levels in colonic samples from both the control and colon cancer groups of AOM/DSS model mice (adapted from Wang et al. (2021) [57] and (2024) [58]). (**a**) The brown color indicates specific immunostaining for HMGB1 in the nuclei and cytoplasm of epithelial cells in colon tissue. Nuclei are not stained with hematoxylin. (**b**) The brown color indicates specific immunostaining for RAGE in the membrane and cytoplasm of epithelial cells in colon tissue. Nuclei are stained with hematoxylin. Original magnification: 100×. Arrows indicate cancer tissue.

**Figure 3 ijms-26-03352-f003:**
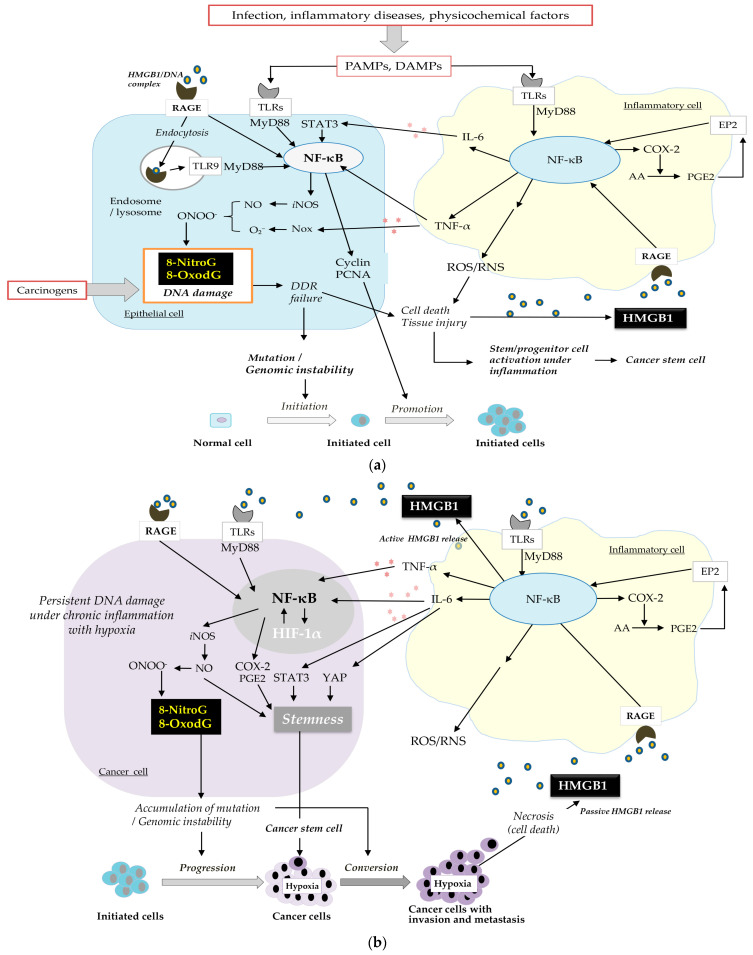
A vicious cycle of DNA damage and inflammation via HMGB1. (**a**) Initiation and promotion, (**b**) progression, and conversion through multiple steps of carcinogenesis.

**Table 1 ijms-26-03352-t001:** Comparison of biomarkers detected by IHC.

Agents	Ref.	Samples	8-NitroG	8-OxodG	PCNA	p16	p53	HIF-1α
OV	[6]	CCA tissues vs adjacent non-cancer tissues	**	**	–	–	–	–
*H. pylori*	[14]	Gastritis with HP vs without HP	**	***	**	–	–	–
	[16]	Gastritis with HP before/after eradication	**	*	–	–	–	–
HCV	[17]	Hepatitis C before/after INF (responder)	*	*	–	–	–	–
HPV	[21]	CIN1 vs condyloma	*	ns	**	ns	–	–
Leukoplakia	[31]	Leukoplakia vs normal mucosa	***	***	ns	–	*	*
UC	[34]	UCAC samples vs UC samples	****	**	–	–	–	–
BE	[36]	BEA samples vs BE samples	*	*	–	–	–	–
		BE samples before/after PPI	**	**	–	–	–	–
MFH	[38]	Samples from deseased vs living MFH	***	–	–	–	–	*

****; *p* < 0.0001, ***; *p* < 0.001, **; *p* < 0.01, *; *p* < 0.05, ns; no significant difference, –; not detected.

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
