# Peer review of "Cancer Development and Progression Through a Vicious Cycle of DNA Damage and Inflammation"

_ijms, 2025, doi:10.3390/ijms26073352_

Round 1
Reviewer 1 Report
Comments and Suggestions for Authors
This review comprehensively explores the interplay between chronic inflammation, DNA damage (particularly 8-nitroG and 8-oxodG), and HMGB1-mediated pathways in cancer development and progression. The authors provide extensive evidence from human and animal studies across multiple cancer types, highlighting 8-nitroG as a potential biomarker and HMGB1 as a central regulator of the inflammation-DNA damage cycle. While the topic is timely and the mechanistic framework is innovative, several critical issues—including structural clarity, methodological transparency, and overinterpretation of data—diminish the manuscript’s impact. Major revisions are required to strengthen the scientific rigor and readability.
Specific Problems and Suggestions for Improvement
- Reorganize Sections 5.1–5.4 into a unified mechanistic diagram or flowchart (beyond Figure 4) that integrates all components (HMGB1, HIF-1α, NF-κB, STAT3, cytokines, DNA damage, and CSCs). Explicitly define how each factor perpetuates the cycle at specific carcinogenic stages (initiation, promotion, progression).
- The manuscript heavily emphasizes 8-nitroG as a "promising biomarker" for diverse cancers (e.g., Page 1, Lines 20–22; Page 3, Lines 104–107; Page 6, Lines 231–234). However, the cited studies are largely observational (e.g., immunohistochemistry in human tissues), with limited validation of its specificity, sensitivity, or clinical utility compared to established biomarkers (e.g., p16 in cervical cancer). Temper claims about 8-nitroG’s diagnostic/prognostic value. Add a dedicated subsection comparing 8-nitroG with other biomarkers (e.g., p16, PCNA) across cancer types, including limitations (e.g., technical challenges in quantification, variability across studies).
While infections (e.g., Opisthorchis viverrini, HPV) are discussed as drivers of inflammation and DNA damage, the role of host factors (e.g., genetic susceptibility, microbiome, immune status) is overlooked. For example, Page 2, Lines 51–56 attributes CSC formation solely to inflammation without addressing genetic/epigenetic contributions. Expand the discussion to include host-microbe interactions, genetic polymorphisms (e.g., iNOS variants), and immune evasion mechanisms that may modulate the inflammation-DNA damage axis.
- Key experimental details are missing in cited studies. For instance, Page 4, Lines 137–139 describes interferon therapy reducing 8-nitroG in hepatitis C patients but omits sample size, follow-up duration, and statistical methods. Similarly, animal studies (e.g., Page 5, Lines 205–208) lack dosage, exposure protocols, and control groups. Include a table summarizing methodologies (e.g., sample sizes, experimental models, detection techniques) of major cited studies to enhance reproducibility and critical assessment.
- Sections 2 ("DNA Damage in Infection-Related Cancer") and 3 ("DNA Damage in Neoplasms Associated with Chronic Inflammatory Conditions") redundantly discuss similar mechanisms (e.g., iNOS/NF-κB activation, 8-nitroG formation). For example, H. pylori-induced gastric cancer (Page 3) and asbestos-related lung cancer (Page 6) both emphasize oxidative/nitrative stress without distinct thematic separation. Merge Sections 2 and 3 into a single section organized by etiological categories (e.g., pathogens, environmental agents, autoimmune disorders) to avoid repetition and improve flow.
- The statement that "PPIs, antibacterial, antiviral, and antiparasitic drugs dramatically diminished DNA lesion markers" (Page 6, Lines 233–236) is overstated. While some studies (e.g., Thanan et al. 2012) show reduced 8-nitroG after PPI treatment, clinical evidence linking this to cancer prevention is lacking. Clarify that these findings are preclinical and highlight the need for longitudinal human studies to validate therapeutic strategies targeting 8-nitroG or HMGB1.
- Figure 2 (UV-exposed mouse skin) lacks quantitative data (e.g., % of stained cells, statistical significance) and uses vague descriptors like "moderately" and "strongly." Figure 3 (HMGB1/RAGE staining) lacks scale bars and negative controls. Revise figures to include quantitative analyses (e.g., histograms, p-values), scale bars, and control samples. Provide high-resolution images with clear labels in the supplemental material.
- The review predominantly supports the authors’ hypothesis but neglects conflicting evidence. For example, some studies suggest HMGB1 has tumor-suppressive roles in early carcinogenesis, which is not addressed (Page 11, Lines 421–423). Acknowledge contradictory findings (e.g., dual roles of HMGB1, context-dependent effects of NF-κB) and discuss how these might refine the proposed model.
This review addresses a critical topic in cancer biology but requires substantial revisions to address structural redundancies, methodological gaps, and overstated claims. By clarifying the mechanistic model, incorporating conflicting evidence, and improving data presentation, the manuscript could become a valuable resource for understanding inflammation-driven carcinogenesis.
Comments on the Quality of English LanguageThe English could be improved to more clearly express the research.
Author Response
Reply to reviewers
We deeply appreciate your and the Reviewers' critical reading of our manuscript, and advices from Assistant Editor, Ms. Gianna Wang. According to the Reviewer's comments, we extensively revised our manuscript. We would like to answer the comments proposed by the Reviewers, and explain the revised version in detail (the changed parts were written in red in the marked manuscript.
First of all, according to advices from Ms. Wang, Figure 1 (original version) was deteled, since the figure concainted a new, unpubshished data, which is described in the Instructions of Authors. Figure 2 and Figure 3 (original version) were modified figures from our published paper, and we have gotten the permission from the publishers, as attached. Also, to reduce self-citation and duplication, we deleted several references and rephrased sentences. We already had native English speakers proofread our English writing (see certification). However, if the English should be improved to more clearly express the research, we will ask Aurhor Services Language Editing, after the acceptance.
Reviewer comments:
Reviwer1
This review comprehensively explores the interplay between chronic inflammation, DNA damage (particularly 8-nitroG and 8-oxodG), and HMGB1-mediated pathways in cancer development and progression. The authors provide extensive evidence from human and animal studies across multiple cancer types, highlighting 8-nitroG as a potential biomarker and HMGB1 as a central regulator of the inflammation-DNA damage cycle. While the topic is timely and the mechanistic framework is innovative, several critical issues—including structural clarity, methodological transparency, and overinterpretation of data—diminish the manuscript’s impact. Major revisions are required to strengthen the scientific rigor and readability.
Reply: We are very grateful for your valuable comments.
Specific Problems and Suggestions for Improvement
- Reorganize Sections 5.1–5.4 into a unified mechanistic diagram or flowchart (beyond Figure 4) that integrates all components (HMGB1, HIF-1α, NF-κB, STAT3, cytokines, DNA damage, and CSCs). Explicitly define how each factor perpetuates the cycle at specific carcinogenic stages (initiation, promotion, progression).
Reply: We newly made Graphic Abstract (in revised version) to show how each factor interact. Sections 5.1–5.4 were reorganized into two sections. Most of 5.1 HIF-1α and 5.2 Factors inducing CSCs were moved into 5.4. Cancer development and progression occur through a vicious cycle between DNA damage and inflammation via HMGB1 (4.2 in revised version).
- The manuscript heavily emphasizes 8-nitroG as a "promising biomarker" for diverse cancers (e.g., Page 1, Lines 20–22; Page 3, Lines 104–107; Page 6, Lines 231–234). However, the cited studies are largely observational (e.g., immunohistochemistry in human tissues), with limited validation of its specificity, sensitivity, or clinical utility compared to established biomarkers (e.g., p16 in cervical cancer). Temper claims about 8-nitroG’s diagnostic/prognostic value. Add a dedicated subsection comparing 8-nitroG with other biomarkers (e.g., p16, PCNA) across cancer types, including limitations (e.g., technical challenges in quantification, variability across studies).
Reply: We belibe that 8-nitroG is a biomarker candidate of inflammation-related carcinogenesis. However, as you pointed out, there are few clinical applications to evaluate it. We tempered the several sentences abou 8-nitroG’s diagnostic/prognostic value.
According to your comment, we add a table (Table 1. Comparison of biomarker detedted by IHC in revised version) about the comprision of biomarkers (8-nitroG and others) .
- While infections (e.g., Opisthorchis viverrini, HPV) are discussed as drivers of inflammation and DNA damage, the role of host factors (e.g., genetic susceptibility, microbiome, immune status) is overlooked. For example, Page 2, Lines 51–56 attributes CSC formation solely to inflammation without addressing genetic/epigenetic contributions. Expand the discussion to include host-microbe interactions, genetic polymorphisms (e.g., iNOS variants), and immune evasion mechanisms that may modulate the inflammation-DNA damage axis.
Reply: We added the sentences about genetic/epigenetic contributions under inflammatory condition, host-microbe interactions, genetic polymorphisms, and immune evasion mechanisms.
“Under inflammatory condition, genetic/epigenetic alterations occur in normal cells, contributing to driver gene mutation, promoter methylation of tumor suppressor genes, and microRNA dysregulation, etc. [2]. Genetic variations influence an individual's vulnerability to stress induced by carcinogens. An imbalance in the microbiota can heighten the likelihood of cancer, particularly colorectal cancer. Mechanisms that allow tumors to evade the immune system may also impact the relationship between in-flammation and DNA damage. While there are other potential ways that interactions between the host and microbes, genetic variations, and immune evasion strategies could affect this relationship, our focus is on how inflammation-driven carcinogenesis begins with DNA damage, notably 8-nitroG, and progresses through a vicious cycle involving HMGB1.” (Lines 48-57)
- Key experimental details are missing in cited studies. For instance, Page 4, Lines 137–139 describes interferon therapy reducing 8-nitroG in hepatitis C patients but omits sample size, follow-up duration, and statistical methods. Similarly, animal studies (e.g., Page 5, Lines 205–208) lack dosage, exposure protocols, and control groups. Include a table summarizing methodologies (e.g., sample sizes, experimental models, detection techniques) of major cited studies to enhance reproducibility and critical assessment.
Reply: To reduce self-citation rate, we deleted several our studies(mainly 2.1.a. Opisthorchis ververrini, 2.2b. Asbestosis and namomaterials in revised version) , and made a table summarizing methods of major cited studies, as supplementary Table 1.
- Sections 2 ("DNA Damage in Infection-Related Cancer") and 3 ("DNA Damage in Neoplasms Associated with Chronic Inflammatory Conditions") redundantly discuss similar mechanisms (e.g., iNOS/NF-κB activation, 8-nitroG formation). For example, H. pylori-induced gastric cancer (Page 3) and asbestos-related lung cancer (Page 6) both emphasize oxidative/nitrative stress without distinct thematic separation.Merge Sections 2 and 3 into a single section organized by etiological categories (e.g., pathogens, environmental agents, autoimmune disorders) to avoid repetition and improve flow.
Reply: We merged Sections 2 and 3 into a single section organized by etiological categories (e.g., pathogens, environmental agents, inflammation-related disorders) to avoid repetition and improve flow.
- The statement that "PPIs, antibacterial, antiviral, and antiparasitic drugs dramatically diminished DNA lesion markers" (Page 6, Lines 233–236) is overstated. While some studies (e.g., Thanan et al. 2012) show reduced 8-nitroG after PPI treatment, clinical evidence linking this to cancer prevention is lacking. Clarify that these findings are preclinical and highlight the need for longitudinal human studies to validate therapeutic strategies targeting 8-nitroG or HMGB1.
Reply: The findings are preclinical studies. As we did not performed longitudinal human studies to validate therapeutic strategies targeting 8-nitroG or HMGB1, we deleted these sentences.
- Figure 2 (UV-exposed mouse skin) lacks quantitative data (e.g., % of stained cells, statistical significance) and uses vague descriptors like "moderately" and "strongly." Figure 3 (HMGB1/RAGE staining) lacks scale bars and negative controls. Revise figures to include quantitative analyses (e.g., histograms, p-values), scale bars, and control samples. Provide high-resolution images with clear labels in the supplemental material.
Reply: Figure 2 and Figure 3 (original version) is the modified figure from our published paper. Therefore, we changed the figure legend to indicate the citation and the permission from publisher, but we cannot add other data.
- The review predominantly supports the authors’ hypothesis but neglects conflicting evidence. For example, some studies suggest HMGB1 has tumor-suppressive roles in early carcinogenesis, which is not addressed (Page 11, Lines 421–423). Acknowledge contradictory findings (e.g., dual roles of HMGB1, context-dependent effects of NF-κB) and discuss how these might refine the proposed model.
Reply: We added the sentences about dual roles of HMGB1 and context-dependent effects of NF-κB. “HMGB1 has some anticancer activities by promoting dendric cells maturation and acti-vating RAGE/NF-kB pathway to polarize M1-like macrophages [53]. “ (Lines 403-405).
This review addresses a critical topic in cancer biology but requires substantial revisions to address structural redundancies, methodological gaps, and overstated claims. By clarifying the mechanistic model, incorporating conflicting evidence, and improving data presentation, the manuscript could become a valuable resource for understanding inflammation-driven carcinogenesis.
Reviewer 2 Report
Comments and Suggestions for Authors
In this review manuscript entitled, “Cancer development and progression through a vicious cycle between DNA damage and inflammation”, the author explored the role of inflammation mediated DNA damage in cancer development and progression.
While this manuscript provides a wide range of interesting information, including inflammation driven 8-NitroG as cancer marker in various cancer models, high mobility group box 1 (HMGB1) protein, HIF1a, NF-KB/iNOS, stat3 pathways; however, this manuscript as a whole lacks cohesion in sentences and paragraphs, counterintuitive statements, incorrect word selections, etc., which needs to be corrected.
Major comments:
- Lines 66-69> Statements are counterintuitive, e.g., 8-nitroG and 8-oxodG formation and iNOS expression appeared to increase in the epithelium of bile duct…………….after a decrease in inflammatory cells. Can the author describe how iNOS expression was increased when inflammatory cells were decreased? It’s thought that inflammatory cells are the major producer of iNOS.
- Lines 71-74> Needs to be rephrased
- Lines 75-78> These lines as a whole need to be rephrased. Can the author describe what was the level of 8-oxodG in CCA patients after praziquantel treatment?
- Why, in some cases, 8-oxodG serves as a biomarker, while in other cases, 8-nitroG is a biomarker?
- Line 131>It would be better in the interest of readers if the author added a few words on how the formation of 8-nitroG and 8-oxodG was measured.
- Line 410> The word “problems” seems incorrect. The author should be specific here. Perhaps they want to say, “tumorigenic DNA lesions”.
- Lines 436-438> No cohesiveness with the previous paragraph.
- Can the author describe the intercommunication among HIF-1a, Nf-kB, and stat3?
Minor comments:
- Line 39> Replace the word “altering” with the word “influencing”
Comments on the Quality of English Language
English needs improvement.
Author Response
Reply to reviewers
We deeply appreciate your and the Reviewers' critical reading of our manuscript, and advices from Assistant Editor, Ms. Gianna Wang. According to the Reviewer's comments, we extensively revised our manuscript. We would like to answer the comments proposed by the Reviewers, and explain the revised version in detail (the changed parts were written in red in the marked manuscript.
First of all, according to advices from Ms. Wang, Figure 1 (original version) was deteled, since the figure concainted a new, unpubshished data, which is described in the Instructions of Authors. Figure 2 and Figure 3 (original version) were modified figures from our published paper, and we have gotten the permission from the publishers, as attached. Also, to reduce self-citation and duplication, we deleted several references and rephrased sentences. We already had native English speakers proofread our English writing (see certification). However, if the English should be improved to more clearly express the research, we will ask Aurhor Services Language Editing, after the acceptance.
Reviewer comments:
Reviewer2
In this review manuscript entitled, “Cancer development and progression through a vicious cycle between DNA damage and inflammation”, the author explored the role of inflammation mediated DNA damage in cancer development and progression.
While this manuscript provides a wide range of interesting information, including inflammation driven 8-NitroG as cancer marker in various cancer models, high mobility group box 1 (HMGB1) protein, HIF1a, NF-KB/iNOS, stat3 pathways; however, this manuscript as a whole lacks cohesion in sentences and paragraphs, counterintuitive statements, incorrect word selections, etc., which needs to be corrected.
Reply: We are very grateful for your valuable comments.
Major comments:
- Lines 66-69> Statements are counterintuitive, e.g., 8-nitroG and 8-oxodG formation and iNOS expression appeared to increase in the epithelium of bile duct…………….after a decrease in inflammatory cells. Can the author describe how iNOS expression was increased when inflammatory cells were decreased? It’s thought that inflammatory cells are the major producer of iNOS.
Reply: The formation of 8-nitroG and 8-oxodG accumulated in the bile duct epithelial cells with repeated OV infection. Inflammatory cells are recruited in acute phase, and decreased thereafter. On the other hand, iNOS expression is continued in the bile duct epithelial cells, since TLRs, which participate the iNOS induction, are expressed in only not immune cells but also epithelial cells. (Lines 69-73)
- Lines 71-74> Needs to be rephrased.
Reply: We deleted and rephrased them to avoid high self-citation and duplication.
- Lines 75-78> These lines as a whole need to be rephrased. Can the author describe what was the level of 8-oxodG in CCA patients after praziquantel treatment?
Reply: The paper indicated the urinary8-oxodG level; Healthy subjects: 3.03 µg/g creatinine, OV infected patients: 4.45 µg/g creatinine, CCA patients 6.83 µg/g creatinine. However, after treatment, data are just shown in a graph, showing about 3.7 µg/g creatinine and 3.2 µg/g creatinine, at 2 months and 1 year, respectively, after praziquantel treatment of OV-infected patients. Since we focus to 8-nitroG and avoid self-citation, we delete this study from the cited references.
- Why, in some cases, 8-oxodG serves as a biomarker, while in other cases, 8-nitroG is a biomarker?
Reply: We think both 8-oxodG and 8-nitroG are formed in inflammation-related cancers. However, 8-NitroG is more specific to inflammation, and therefore we would like to focus to 8-nitroG, as a biomarker, in the revised manuscript.
- Line 131>It would be better in the interest of readers if the author added a few words on how the formation of 8-nitroG and 8-oxodG was measured.
Reply: We added immunohistochemistry (IHC) as the method to detect 8-nitroG and 8-oxodG.
- Line 410> The word “problems” seems incorrect. The author should be specific here. Perhaps they want to say, “tumorigenic DNA lesions”.
Reply: Thank you. We changed it to ”tumorigenic DNA lesions”.
- Lines 436-438> No cohesiveness with the previous paragraph.
Reply: We deleted the sentences.
- Can the author describe the intercommunication among HIF-1α, Nf-kB, and stat3?
Reply: The intercommunication among HIF-1α, Nf-kB, and stat3 is shown in Graphic Abstract (in revised version), which we newly made.
- Minor comments:
Line 39> Replace the word “altering” with the word “influencing”
Reply: We changed the word.
Round 2
Reviewer 1 Report
Comments and Suggestions for Authors
I have no further questions.